# Sirtuin 2 Dysregulates Autophagy in High-Fat-Exposed Immune-Tolerant Macrophages

**DOI:** 10.3390/cells10040731

**Published:** 2021-03-26

**Authors:** Sanjoy Roychowdhury, Anugraha Gandhirajan, Christopher Kibler, Xianfeng Wang, Vidula Vachharajani

**Affiliations:** 1Department of Inflammation and Immunity, Lerner Research Institute, Cleveland, OH 44195, USA; roychos@ccf.org (S.R.); GANDHIA3@ccf.org (A.G.); KIBLERC@ccf.org (C.K.); 2Department of Medicine, Wake Forest School of Medicine, Winston-Salem, NC 27101, USA; xwang@wakehealth.edu; 3Department of Critical Care, Respiratory Institute, Cleveland Clinic, Cleveland, OH 44195, USA

**Keywords:** sepsis, SIRT2, SIRT, obesity, autophagy

## Abstract

**Simple Summary:**

SIRT2 regulates autophagy in high-fat-exposed immune cells.

**Abstract:**

Obesity increases morbidity and resource utilization in sepsis patients. The immune response in sepsis transitions from an endotoxin-responsive hyper- to an endotoxin-tolerant hypo-inflammatory phase. The majority of sepsis mortality occurs during hypo-inflammation. We reported prolonged hypo-inflammation with increased sirtuin 2 (SIRT2) expression in obese-septic mice. The effect of direct exposure to high-fat/free fatty acid (FFA) and the role of SIRT2 in immune cells during the transition to hypo-inflammation is not well-understood. Autophagy, a degradation process of damaged protein/organelles, is dysregulated during sepsis. Here, we investigated the effect of direct FFA exposure and the role of SIRT2 expression on autophagy as macrophages transition from hyper-to hypo-inflammation. We found, FFA-exposed RAW 264.7 cells with lipopolysaccharide (LPS) stimulation undergo endotoxin-sensitive (“sensitive”) hyper- followed by endotoxin tolerant (“tolerant”) hypo-inflammatory phases; SIRT2 expression increases significantly in tolerant cells. Autophagy proteins LC3b-II, and beclin-1 increase in FFA-sensitive and decrease in tolerant cells; p62 expressions continue to accumulate in tolerant cells. We observed that SIRT2 directly deacetylates α-tubulin and impairs autophagy clearance. Importantly, we find SIRT2 inhibitor AK-7 treatment during endotoxin tolerant phase reverses autophagy dysregulation with improved autophagy clearance in FFA-tolerant cells. Thus, we report impaired autophagosome formation and autophagy clearance via increased SIRT2 expression in FFA-exposed tolerant macrophages.

## 1. Introduction

The global burden of sepsis is rising, with approximately 45 million new cases of sepsis and 11 million deaths reported in 2017 [1]. In the US alone, sepsis kills over 270,000 patients each year [2], with inpatient mortality of 60% for septic shock and 36% for severe sepsis [3]. Medicare cost of inpatient care for sepsis rose from $27.7 billion in 2012 to $41.5 billion in 2018 [3]. Currently, there are no sepsis-specific therapies in use [4]. Immune response in sepsis transitions from early/hyper-inflammatory to a late/hypo-inflammatory and immunosuppressive phase [5,6]; the majority of sepsis-related mortality occurs during hypo-inflammation with multiple organ dysfunction syndromes [7]. The hypo-inflammatory phase of sepsis is characterized by endotoxin tolerance [5,8].

The obesity epidemic, affecting over one in three adults in the United States, increases morbidity and resource utilization in critically ill patients, including sepsis [9]. Obesity is an independent risk factor for increased mortality in viral-sepsis of COVID-19 [10]. We reported that obesity exaggerates hyper- and prolongs the hypo-inflammatory phase in the microvasculature in septic mice [11]. The effect of high-fat and free fatty acid (FFA) exposure on the innate immune response to an acute inflammatory stimulus is emerging. While high-fat diet intake exaggerates pro-inflammatory cytokine expression during the early/hyper-inflammation [12] and decreases it during late sepsis/hypo-inflammation [13], the mechanisms are not well understood. Since the majority of sepsis-related deaths occur during hypo-inflammation [7], it is critical to assess the effect of high-fat exposure on immune response during the hyper- and hypo-inflammatory phases of sepsis.

Sirtuins, a highly conserved family of nutrient sensors, are known for their anti-inflammatory and anti-oxidant properties. The seven members of mammalian homologs of SIRTs, SIRT1-7, are dispersed through the cell compartments. SIRTs 1, 6 and 7 are nuclear SIRTs 3, 4, and 5 mitochondrial, and SIRT2 is cytosolic [14]. We showed that the SIRTs are crucial in the transition from hyper- to hypo-inflammation and sustenance of hypo-inflammation with sepsis [5,11]. Specifically, SIRT2 expression is critical for prolonged hypo-inflammation in obese mice with sepsis, and SIRT2 inhibition in ob/ob mice during the hypo-inflammatory phase of sepsis reverses hypo-inflammation and improves survival [11].

Autophagy, a process of controlled degradation of damaged organelles, proteins and engulfed pathogens, is a crucial mechanism for cell survival in sepsis [15]. Evidence suggests that autophagy is induced during early sepsis and dysregulated during late/hypo-inflammation with organ dysfunction in various cell types and linked to impaired pathogen clearance [15,16], but the exact mechanisms are unclear. Obesity induces autophagy in adipose tissue, macrophages and hepatocytes [17,18,19,20], but the effect of high-fat exposure on autophagy in immune cells in sepsis is not well understood. Lastly, while SIRT2 affects autophagy in various cell types affecting neurodegenerative diseases [21,22], the role of autophagy in the innate immune cells with high-fat exposure is unknown. In this project, we aimed to study the role of SIRT2 in modulating autophagy along with inflammatory response in free fatty acid-exposed macrophages using macrophage cell line in vitro during hyper and hypo-inflammatory phases.

## 2. Materials and Methods

Antibodies: Antibodies were obtained from the following vendors: SIRT2 (Cell signaling Technology, Danvers, MA, USA D4050), LC3b-II, (Cell signaling Technology E5Q2K), p62 (Abnova, Taipei, Taiwan, 2C11), beclin-1 (Cell signaling, D40C5), total tubulin (Sigma, St. Louis, MO, USA T6199), acetyl tubulin (Cell signaling, D20G3), GAPDH (Santa Cruz Biotechnology, Heidelberg, Germany, sc-322336C5); anti-mouse IgG, HRP-linked antibody (Cell signaling, 7076), anti-rabbit IgG, HRP-linked antibody (Cell signaling, 7074)) Alexa Fluor 488 and 555 from Invitrogen (Carlsbad, CA, USA), biotinylated human SIRT2 antibody (R&D system, Minneapolis, MN, USA, BAF4358). Chemicals were purchased from the following vendors: AK-7 from TOCRIS Bioscience (Minneapolis, MN, USA), Chloroquine diphosphate (P36239, Life technologies, Carlsbad, CA, USA), rapamycin, fatty-acid-free bovine serum albumin, stearic acid and lipopolysaccharide (LPS) from Sigma-Aldrich (St. Louis, MO, USA). TNF-α and IL6 ELISA kit was obtained from BioLegend (San Diego, CA, USA), and an autophagy flux assay kit was purchased from Invitrogen (Carlsbad, CA, USA).

### 2.1. RAW 264.7 Cells with Stearic (Free Fatty Acid: FFA) Acid Exposure

RAW 264.7 cell macrophages (RAW cells) were purchased from ATCC. RAW cells were pretreated with bovine serum albumin (BSA) (vehicle) or stearic acid (free fatty acid: FFA) (final concentration 200 µM) overnight as used in our previous publication, followed by further treatment with the LPS (final 100 ng/mL) [11,12]. We did not find evidence of FFA-induced toxicity in RAW cells. We studied immune and autophagy response in cells with either 4 h or 24 h of the LPS stimulation. To test for autophagy and immune response with endotoxin tolerance, we re-stimulated cells with a second dose of the LPS (100 ng/mL) for 4 h. Autophagy proteins were assessed in the cytoplasmic fraction discussed below. Total RNA was isolated for real-time PCR to detect TNF-α mRNA expression level [11,12,23]. To study the SIRT2 expression profile in RAW, macrophages were pretreated with BSA (vehicle) or FFA (final concentration 200 µM) overnight, followed by further treatment with the LPS (final 100 ng/mL) for 0, 2, 4, 6, or 24 h, respectively. Total protein was collected for Western blotting. The band signals of SIRT2 and GAPDH were quantified to plot [11,12,23]. To study the effect of chloroquine on autophagy inhibition in FFA-treated RAW cells, we cultured the cells in Petri dishes the previous day of treatment. Cells were treated with chloroquine (90 µM) along with FFA and induced with the LPS, as mentioned previously. At the end of the incubation, cells were processed, cytoplasmic fractions were isolated and analyzed for LC3B protein expression by Western blot.

### 2.2. Cytoplasmic Extraction from RAW Cells

Cytoplasmic fractions were prepared using a mitochondria isolation kit for cultured cells (from Thermo Scientific, Rockford, IL, USA). Briefly, RAW cells were cultured in Petri dishes (approximately 20 × 10^6^ cells per plate) the previous day of treatment. Cells were treated and induced as indicated above. Posttreatment, cells were scraped in ice-cold PBS and harvested by centrifugation at 850× *g* for 2 min. Cells were processed, and the cytoplasmic extractions were carried out as per the manufacturer’s instructions. We studied beclin-1, LC3b, p62, acetylated α-tubulin and α-tubulin expression (as indicated in figures) using the Western blot method in the cytoplasmic fraction.

### 2.3. Immunocytochemistry

RAW264.7 cells were plated on 8-chambered Lab-Tek slides. After culturing for 3 h, once cells were attached firmly with the plate, cells were exposed to bovine serum albumin (5% BSA) or stearic acid–BSA conjugate (final concentration 100 µM) in the presence or absence of lipopolysaccharide (LPS, 100 ng/mL). For SIRT2 inhibitor treatment, cells were treated with AK-7 4 h after the 1st LPS exposure. For the second LPS exposure, LPS were added to the media 20 h after the first LPS exposure. At the end of the incubation (as indicated in figures), cells were fixed using 4% paraformaldehyde, washed with phosphate-buffered saline (PBS) and permeabilized with 0.1% Triton-X-100 for 10 minutes. Cells were then washed with PBS, blocked for 1 hour with 5% BSA (assay diluent) and incubated overnight with primary antibodies diluted in assay diluent at 4 °C. The next day, cells were washed with PBS and incubated in the dark with respective secondary antibodies diluted in assay diluent (1:500) for 2 h at room temperature. Following 3 further PBS wash, slides were mounted with a nuclear fluorophore (DAPI)-containing mounting media. Images were acquired using a Leica confocal microscope (Leica Microsystems, Buffalo Grove, IL, USA) using 63 × objectives. Images were semi-quantified using Image Pro-Plus software (Media Cybernetics, Bethesda, MD, USA).

### 2.4. Autophagy Flux Assay

RAW264.7 cells were plated on 8-chambered Lab-Tek slides and exposed to different treatments as indicated above. Cells were transduced with an mRFP-GFP-LC3b-II construct according to the manufacturer’s instruction [24]. A separate group of cells was also treated with rapamycin (100 ng/mL, 24 h). At the end of the incubation (as indicated in figures), cells were fixed using 4% paraformaldehyde, washed with phosphate-buffered saline (PBS) and mounted with a DAPI-containing media. Images were acquired as described above.

### 2.5. Immunoprecipitation Assay

For immunoprecipitation, RAW cells were treated with FFA and induced with the LPS, as mentioned previously. After post-treatment, cells were washed with PBS and were resuspended in NP-40 lysis buffer (containing 1% Nonidet *P*-40, 50 mM Tris (pH 8), 150 mM NaCl) along with phosphatase and protease inhibitors mixture (Roche Diagnostics, Mannheim, Germany). Then cells were lysed using a sonicator for 5 × 10 s with a 20 s interval between each sonication at a frequency of 10 kHz. Cell debris was removed by centrifugation at 10,000× *g* for 15 min, and the supernatant was treated with biotinylated anti-SIRT2 antibody for SIRT2 pull-down or rabbit IgG control antibody as the negative control. SIRT2 and biotinylated antibody complexes were immunoprecipitated using streptavidin magnetic beads (Life technologies, Carlsbad, CA, USA). Beads were extensively washed 3 times with washing buffer. After the final wash, the supernatant was discarded, and the pellet was resuspended in 100 µL of 2× sample buffer with BME and denatured at 95 °C for 10 min. Denatured precipitates were subjected to SDS–PAGE (12% gel) followed by transfer to 0.2 μm PVDF membrane.

### 2.6. Western Blot Analysis

Both cytoplasmic and immunoprecipitated samples were subjected to SDS–PAGE (4–15% gel) followed by transfer into 0.2 µm pore size polyvinylidene difluoride (PVDF) membrane. 40 µg of protein was loaded per lane in all the gels. The membrane was blocked with 5% skimmed milk in Tris-buffered saline TBS with tween (0.05%) (TBST) for 90 min at room temperature. Blots were incubated with primary antibodies to analyze the autophagic protein expression overnight at 4 °C. The blots were washed thrice with TBST and incubated with anti-rabbit/mouse IgG, HRP-linked secondary antibody for 1 h at room temperature. ECL (Bio-Rad, Hercules, CA, USA) was used for detection, and images were captured using the Chemi Doc imaging system (Bio-Rad, Hercules, CA, USA) and were quantified using Image J software (NIH, Bethesda, MD, USA)

### 2.7. ELISA

RAW264.7 cells were plated on 24 well plates. After culturing for 3 h, once the cells attached firmly with the plate, they were exposed to bovine serum albumin (5% BSA) or stearic acid–BSA conjugate (final concentration 100 µM) in the presence or absence of lipopolysaccharide (LPS, 100 ng/mL, 1st LPS). After 20 h, media were removed from the wells and cells were treated with fresh media containing vehicle or second LPS. At the end of the incubation, TNF-α and IL6 ELISA were performed according to the manufacturer’s instruction. For SIRT2 inhibitor treatment, cells were treated with AK-7 4 h after the 1st LPS exposure.

### 2.8. Statistical Analysis

All data were expressed as the mean ± standard error of the mean (SEM) with n = 3–4 data points per experimental group. Statistical analysis was performed using Prism software version 5.02 (GraphPad Software, San Diego, CA, USA). For comparing multiple groups, analysis of variance was used with a Newman–Keuls post hoc test. Statistical significance was defined as *p* < 0.05. A Student’s t-test was used for the parametric analysis of two groups.

## 3. Results

### 3.1. SIRT2 Expression Is Increased in Tolerant Macrophages

We used mouse RAW 264.7 (RAW cells) macrophage cell line to define the effect of free fatty acid on endotoxin tolerance; we exposed RAW cells to stearic acid-BSA conjugate (FFA) or BSA alone (vehicle: BSA) and stimulated with one or two doses of the LPS based on reported literature [25]. The FFA-exposed RAW cells showed significantly higher TNF-α mRNA in response to 1st LPS (4 h duration) compared to BSA-exposed cells, indicative of exaggerated hyper-inflammation (Sensitive cells). Both BSA- and FFA-exposed macrophages were unable to further induce TNF-α mRNA following the second LPS challenge, indicating endotoxin tolerance (tolerant cells) within 4 h of 1st LPS stimulation (Figure 1A).

Next, we assessed SIRT2 protein expression in endotoxin-sensitive and tolerant RAW cells using Western blot techniques. We stimulated BSA and FFA-exposed cells with the LPS and studied SIRT2 expression at different time points. We observed that the SIRT2 expression (using 40 µg protein/lane to load gel), including two isoforms (43 kD and 39 kD) [26], increased in both BSA and FFA-exposed cells in a time-dependent manner, but was significantly higher in only FFA-exposed endotoxin-tolerant (24 h post-LPS challenge) vs. sensitive cells, as shown in the Western blot images (Figure 1B) and image quantification (Figure 1C). Of note, the SIRT2 expression at 24 h post-LPS stimulation in FFA-exposed cells was significantly higher than BSA-exposed cells.

Using immunocytochemistry, we confirmed SIRT2 expression in FFA-exposed cells. We found that SIRT2 expression, indicated by immunofluorescence measurement, increased with the LPS in sensitive cells (with single LPS stimulation). SIRT2 expression increased in tolerant cells with and without LPS (second LPS) stimulation (representative image: Appendix A; image quantification: Appendix A and Western blot image: Appendix A; Blot quantification: Appendix A).

Further, to test the role of SIRT2 on endotoxin tolerance in FFA-exposed tolerant cells, we treated tolerant cells with SIRT2 inhibitor AK-7 and assessed TNF-α protein expression in the supernatant using ELISA with and without LPS. We observed that the tolerant cells without AK-7 (vehicle) showed a muted response to the LPS stimulation (compared to FFA-sensitive cells), indicating endotoxin tolerance, while AK-7-treated cells showed a robust response to the LPS (Figure 1D) and reversal of endotoxin tolerance. Thus, in agreement with our published work [11], we showed that SIRT2 expression is crucial for endotoxin tolerance in FFA-exposed cells.

We also studied interleukin 6 (IL6) expression in the supernatants from FFA-exposed sensitive cells and tolerant cells with and without AK-7 treatment (Figure 1E). We observed that IL6 increased significantly in sensitive cells vs. control and decreased in tolerant cells vs. sensitive cells. Consistent with the literature, IL6 expression increased in response to the LPS in vehicle-treated tolerant cells [27,28]. AK-7 treatment decreased IL6 in tolerant cells (-LPS) vs. vehicle; the mechanisms need further evaluation. LPS stimulation significantly increased IL6 expression in AK-7 treated FFA-tolerant cells vs. AK-7 alone (-LPS). Next, we studied the role of SIRT2 in autophagy regulation in FFA-exposed cells.

### 3.2. Free Fatty Acid Exposure Dysregulates Autophagy in Tolerant Macrophages

Exposure to free fatty acid is linked to inhibition of autophagy and impaired autophagy flux in a variety of cells [29]. Recent evidence indicates that immunosuppression with endotoxin tolerance is linked to dysregulated autophagy in sepsis [30,31]. We aimed to study the effect of free fatty acid-induced SIRT2 expression on autophagy regulation during hyper- (sensitive) and hypo-inflammatory (tolerant) phases. Autophagy is a multistep process broadly involving autophagosome formation and lysosome fusion to form autolysosome with protein degradation. Several proteins participate in the process. We studied the expression profile of key representative autophagy proteins in FFA-exposed sensitive and tolerant cells.

#### 3.2.1. Effect of SIRT2 Expression on Autophagosome Formation

Beclin-1: Beclin-1 protein is involved in the initiation and elongation of autophagosome [32]. We found a trend towards increased beclin-1 expression in response to the LPS in sensitive cells and significantly decreased in tolerant vs. sensitive cells (representative image: Figure 2A and image quantification: Figure 2B). To elucidate the role of SIRT2 in this process, we studied the effect of SIRT2 inhibition using AK-7 or vehicle on beclin-1 expression in tolerant cells with and without (second) LPS stimulus. We observed increased beclin-1 in AK-7 treated cells with and without LPS stimulation vs. vehicle (representative WB image: Figure 2C and WB quantification: Figure 2D).

Light Chain 3b-II expression: Microtubule-associated protein 1A/1B-light chain 3 (LC3) is a ubiquitously expressed protein in mammalian cells. During autophagy, the cytosolic LC3b (LC3b-I) is conjugated to phosphatidylethanolamine to form LC3b-phosphatidylethanolamine conjugate (LC3b-II), which is crucial for the autophagy-elongation process; LC3b-II is recruited to autophagosomal membranes [33]. Using Western blot assay, we observed that LC3b-II expression increased significantly in FFA-sensitive cells in response to the LPS. In FFA-tolerant cells, LC3b-II expression decreased significantly vs. sensitive cells (representative image: Figure 3A and image quantification: Figure 3B). To elucidate the role of SIRT2 in modulating LC3b-II, we studied the effect of AK-7/vehicle on LC3b-II expression in tolerant cells. We found that LC3b-II expression increased significantly in AK-7 treated tolerant cells with and without second LPS stimulation vs. vehicle (representative image Figure 3C; WB image quantification: Figure 3D).

To confirm the role of LC3B induction, we studied the role of rapamycin, an autophagy inducer, on LC3 expression using immunocytochemistry in FFA-exposed cells with 4 h and 24 h LPS stimulation. We observed, as expected that rapamycin-induced autophagy in FFA-exposed cells. At 4 h post-LPS, while cells were able to clear autophagy (decreased yellow puncta), while at 24 h (tolerant cells), there was a strong trend towards the accumulation of LC3 in the vehicle and rapamycin-treated cells (Representative images: Appendix A and image quantification: Appendix A), confirming dysregulation of autophagy in FFA-tolerant cells. To further study the role of autophagy inhibition in FFA-exposed sensitive and tolerant RAW cells, cells were exposed to chloroquine (CQ, 90 µM, 24 h). CQ inhibits the fusion of the autophagosome with a lysosome. Although CQ exposure increased LC3b-II expression in FFA-exposed control RAW cells (Appendix A), it did not significantly increase LC3b-II in tolerant RAW cells. Since autophagy is already blunted in FFA-exposed tolerant RAW cells, exposure to an additional pharmacological autophagy inhibitor was unable to further increase LC3b-II accumulation in these cells.

Accumulation of p62: p62 is known as an autophagy receptor. We studied p62–expression in FFA-exposed sensitive and tolerant cells. Using Western blot analysis, we observed a significant increase in p62 expression in sensitive cells and continued accumulation of p62 in tolerant cells (representative WB image: Figure 4A and WB image quantification: Figure 4B). We then studied the effect of SIRT2 inhibitor AK-7/vehicle on p62 expression in tolerant cells. We found that p62 expression decreased in AK-7 treated tolerant cells with and without second LPS stimulation vs. vehicle (representative image Figure 4C; WB image quantification: Figure 4D).

As an autophagy receptor, p62 is known to interact with other autophagy proteins to then form an autophagosome. Next, we studied colocalization between LC3b and p62 [34] using immunocytochemistry and observed increased colocalization (yellow puncta) of LC3b (green) and p62 (red) in FFA-exposed sensitive cells and decreased significantly in tolerant cells. Tolerant cells exhibited an increased number of p62-positive red puncta (representative image: Figure 5A and fluorescence-quantification: Figure 5B). This may be a reflection of low LC3b-II and high p62 expression in tolerant cells (Figure 4). We then elucidated the effect of AK-7/vehicle treatment on colocalization in FFA-tolerant cells. We found increased LC3b-p62 colocalization in AK-7 treated FFA-tolerant cells with and without LPS stimulation vs. vehicle exposure (representative image: Figure 5C and immunofluorescence quantification: Figure 5D).

Acetylated α-tubulin (Ac-tubulin) expression in the microtubules is essential to carry autophagosomes to the lysosome to form autolysosomes [35]. SIRT2 is a well-recognized α-tubulin deacetylator [26,36]. Therefore, next, we studied α-tubulin acetylation (Ac-tubulin) in FFA-exposed sensitive and tolerant cells to test the hypothesis that increased SIRT2 inhibits autophagosome- clearance via α-tubulin-deacetylation. We observed that, while Ac-tubulin expression did not change in FFA-sensitive cells, while it decreased significantly in FFA-tolerant cells (representative image: Figure 6A and image quantification: Figure 6B). Furthermore, we observed that Ac-tubulin expression increased in AK-7 treated FFA-tolerant cells with the LPS stimulation vs. vehicle treatment (representative image: Figure 6C and image quantification: Figure 6D). Thus, consistent with the literature, we observed that SIRT2 deacetylates α-tubulin and SIRT2 inhibition increases Ac-tubulin expression [36,37]. Of note, we did not observe the change in total α-tubulin expression in either sensitive versus tolerant cells or in tolerant cells with or without AK-7/second LPS stimulation (Appendix A).

SIRT2 colocalizes with key autophagy proteins: We then studied whether beclin-1, LC3bII, p62 and α-tubulin colocalize with SIRT2 using SIRT2 immunoprecipitation (representative images: Figure 7A and respective inputs: Figure 7B). We observed that, while LC3b-II, p62 and α-tubulin colocalized with SIRT2 in FFA-sensitive and tolerant cells. We did not observe SIRT2 and beclin-1 colocalization (not shown), indicating that the AK-7-effect on increased beclin-1 is not a direct effect of SIRT2 but rather an indirect one. We also confirmed SIRT2 and p62 colocalization using immunocytochemistry (Appendix A).

#### 3.2.2. Autophagy Clearance

To further confirm the impairment in autophagy clearance, we monitored autophagy flux in FFA-exposed RAW cells using a tandem mRFP-GFP-LC3 reporter (Invitrogen) probe coupled with immunocytochemistry. In the presence of this probe, yellow puncta are visible in the cells at the early stages of autophagosome formation. When autophagosomes merge with lysosomes and autolysosomes are formed, green color is quenched due to the acidic pH of the lysosome, and therefore, only red puncta are visible. We observed increased green (LC3) and few yellow puncta, indicating autophagosome formation in sensitive cells vs. control (Figure 8A,B). Furthermore, we observed that the autophagosomes (yellow puncta) continued to accumulate in vehicle-treated-tolerant cells suggesting impaired clearance. However, autophagosome (yellow puncta) significantly decreased in AK-7-treated tolerant cells, indicating autophagy clearance (Figure 8C,D).

In summary, we show that in FFA-exposed macrophages, autophagy is induced during the hyper-inflammatory (sensitive) phase and impaired during the hypo-inflammatory (tolerant) phase in response to the LPS stimulation. Moreover, we show that SIRT2 accumulates and co-immunoprecipitates with autophagy regulator protein LC3b-II and receptor p62. Furthermore, SIRT2 deacetylates and deactivates α-tubulin, which is essential for carrying the autophagosome to the lysosome for autolysosome formation and clearance, thus impeding the clearance [26,35,38]. Importantly, we also show that AK-7 treatment during the hypo-inflammatory phase reverses autophagy-impairment via increasing beclin-1, LC3b-II, α-tubulin acetylation expression required for autophagy activation, and decreasing p62 accumulation as a result of autophagy clearance in tolerant cells.

## 4. Discussion

The main goal of this project was to study the role of SIRT2 in autophagy dysregulation in immune cells with acute exposure to a free fatty acid. Using mouse and cell models, we showed the effect of high-fat-induced obesity on sepsis outcomes in mice previously [11]. We aimed to further elucidate the effect of acute (LPS) on chronic (FFA) inflammation using direct exposure of FFA on macrophages with endotoxin (LPS) stimulation. Consistent with our previous work, SIRT2 expression increased during the hypo-inflammatory phase with endotoxin tolerance in high-fat-exposed RAW cell macrophages; SIRT2 inhibition using AK-7 reversed endotoxin tolerance [25,37]. Furthermore, we show that autophagy, a protective mechanism for cell survival, is induced during the endotoxin-sensitive hyper-inflammatory phase and dysregulated during the endotoxin-tolerant hypo-inflammatory phase, along with increased SIRT2 expression in RAW cell macrophages. Moreover, SIRT2 directly interacts with several of the autophagy regulators, including those integral to autophagy initiation and elongation process (LC3b-II), autophagy receptor (p62) and protein essential for microtubule assembly (α-tubulin) to carry autophagosome to form autolysosome. Specifically, we found impaired autophagy in FFA-exposed tolerant macrophages at three different levels: (1) decreased LC3b-II and beclin-1 [39]; (2) accumulation of p62 [40]; (3) deacetylation of α-tubulin by SIRT2. Acetylation of α-tubulin is essential for microtubule assembly [35]. We found that changes in all four (beclin-1, LC3bII, p62 and α-tubulin) reversed in cells treated with SIRT2 inhibitor AK-7 during the hypo-inflammatory phase. We found a direct interaction of LC3b-II, p62 and α-tubulin with SIRT2, but not beclin-1. The effect of AK-7 on beclin-1 expression without direct interaction between SIRT2 and beclin-1 suggests an indirect effect of SIRT2 on beclin-1 expression. The exact role of SIRT2 in modulating these proteins needs further elucidation.

Decreased LC3b-II-p62 interaction in FFA-tolerant cells suggests impaired autophagosome formation. We find that AK-7 treatment reverses this impaired interaction. Acetylation of α-tubulin is essential for microtubule assembly and function to carry autophagosome to the lysosome for autolysosome formation and degradation [26,37]. We find, consistent with literature that in FFA-tolerant macrophages, α-tubulin is deacetylated and autophagy clearance (autophagy flux assay) is impaired. AK-7 treatment increases acetylated α-tubulin and reverse impaired-autophagy clearance. Together, we show that several of the key autophagy regulators, such as LC3b-II, p62 and α-tubulin directly and beclin-1 indirectly interact with SIRT2.

There are several limitations in this study, however. We studied the role of autophagy in the LPS-induced cells model of RAW macrophages in vitro in this project. The effect of a high-fat diet on autophagy in vivo in mice and ultimately in human tissue/cells needs further evaluation. Furthermore, we studied the cytoplasmic fraction in these experiments since the autophagy process takes place in the cytoplasm, and SIRT2 is mainly a cytoplasmic protein. However, SIRT2 protein translocates to the nucleus as well as mitochondria [22,41]. It is possible that SIRT2 affects the transcription of autophagy proteins in FFA-exposed sensitive and tolerant cells. While out of the scope of the present work, transcriptional regulation of autophagy genes in FFA-exposed cells needs further elucidation in future studies.

During sepsis, immune cells, including macrophages, detect bacterial products or endotoxins in the circulation and trigger inflammation. Although excessive inflammation in tissues triggers cell death, restrained inflammation is essential for pathogen clearance during sepsis. We reported previously that the exaggerated hyper- followed by a prolonged hypo-inflammatory phase with endotoxin tolerance via increased SIRT2 expression in FFA-tolerant cells; evidence suggests endotoxin tolerance is linked to impaired bacterial clearance in mice [11,42]. Moreover, in our previous work, we showed that SIRT2 inhibition using AK-7 reversed in vivo endotoxin tolerance and improved survival in obese mice [11]. In this study, we demonstrated that along with the reversal of endotoxin tolerance, AK-7 also reverses impaired autophagy in macrophages with high-fat exposure.

The cytoprotective role of autophagy is known; furthermore, the role of SIRT2 in autophagy in neurological disorders, including Alzheimer’s disease, is also known [35]. To our knowledge, this is the first report of the role of SIRT2 in the dysregulation of autophagy in the innate immune cell. Moreover, we show that endotoxin-tolerance reversal along with the reversal of autophagy linking the two via SIRT2 for the first time. Although we do not claim that autophagy is the only mechanism for endotoxin tolerance, we feel it is critically important to further elucidate the relationship between the two.

In conclusion, we show that SIRT2 contributes to the development of immune tolerance in vitro by impairing the removal of damaged organelles and pathogens via autophagy dysregulation. Inhibition of SIRT2 during the hypo-inflammatory phase improves autophagy. These data suggest SIRT2 inhibition may be a potential therapeutic target to ameliorate immune dysfunction during the late phase of sepsis.

## Figures and Tables

**Figure 1 cells-10-00731-f001:**
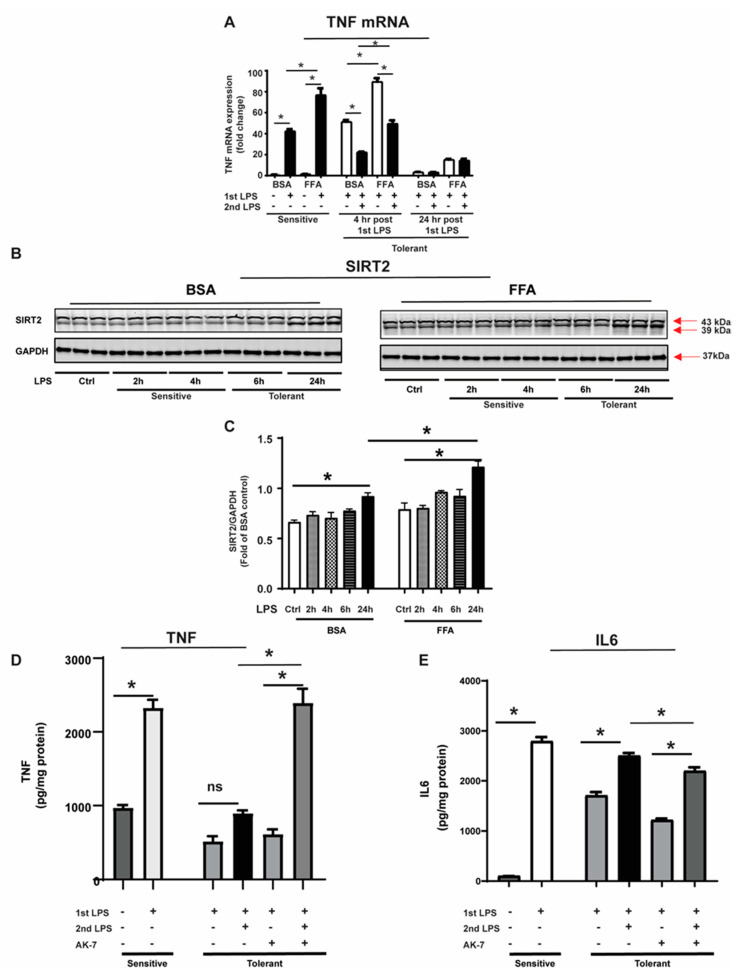
SIRT2 expression is increased in free fatty acid (FFA)-exposed tolerant RAW 264.7 cell macrophages (RAW) cells. Bovine serum albumin (BSA: vehicle for free fatty acid) or stearic acid (free fatty acid: FFA)-exposed RAW264.7 cell macrophages (RAW) were stimulated with either one or two doses of the LPS; second dose at indicated time points after 1st LPS. (**A**) TNF-α mRNA expression was evaluated by qRT–PCR analysis. (**B**) SIRT2 protein expression was detected by Western blot in BSA- or FFA-exposed RAW cells following a single dose of the LPS stimulation for indicated time periods. GAPDH was used as the loading control. (**C**) Western blot image quantification using image-J software (n = 3 each group; * *p* < 0.05). SIRT2 protein expression was normalized to GAPDH. (**D**,**E**). To inhibit SIRT2, after 4 h of 1st LPS exposure, cells were treated with SIRT2 inhibitor AK-7 (25 µM) and incubated further for 20 h in the presence or absence of the second dose of the LPS. TNF-α protein released in the cell culture media from FFA-exposed sensitive and tolerant cells was evaluated by ELISA. (* *p* < 0.05). IL6 (Figure 1E) protein released in the cell culture media from FFA-exposed sensitive and tolerant cells evaluated by ELISA. (* *p* < 0.05).

**Figure 2 cells-10-00731-f002:**
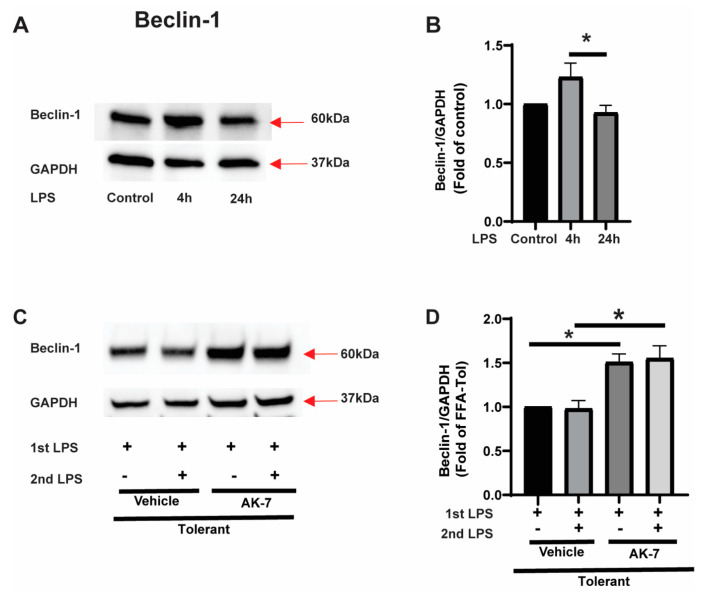
Beclin-1 expression in sensitive and tolerant macrophages. Stearic acid (free fatty acid: FFA)-exposed sensitive and tolerant RAW264.7 cell macrophages (RAW) were stimulated with or without lipopolysaccharide (LPS) for 4 h or 24 h. Loading control: GAPDH. (**A**) Beclin-1 protein expression was detected in the cytosolic extract by Western blot. (**B**) Western blot image quantification using image-J software (n = 5 blots; * *p* < 0.05). (**C**) FFA-exposed tolerant RAW cells were treated with SIRT2 inhibitor AK-7 (25 µM) or vehicle (DMSO; equal volume) and incubated further for 20 h and stimulated with or without LPS as indicated. Beclin-1 protein expression was detected in the cytosolic extract by Western blot (**D**). Western blot image quantification of beclin-1using image-J software (n = 5 blots; * *p* < 0.05).

**Figure 3 cells-10-00731-f003:**
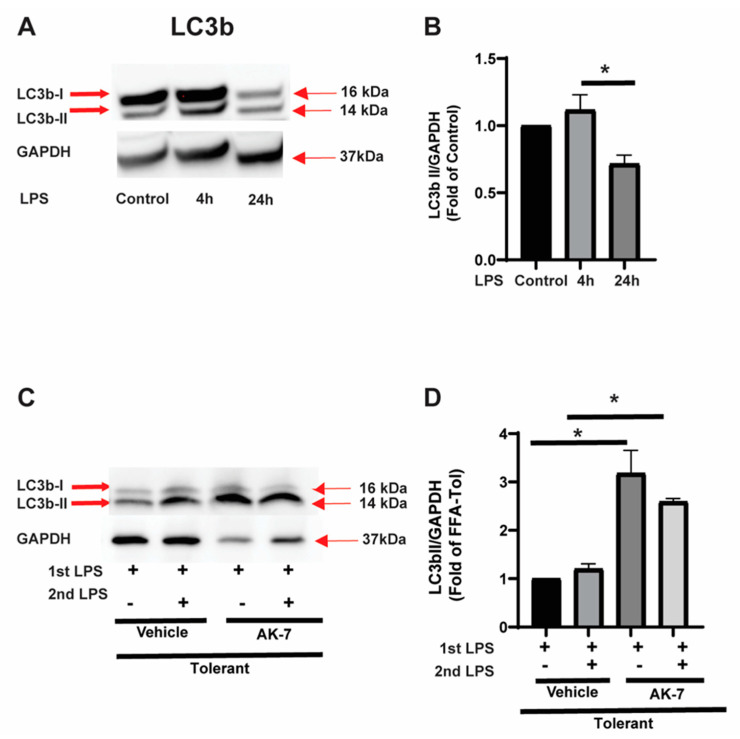
LC3b expression in sensitive and tolerant macrophages. Stearic acid (free fatty acid: FFA)-exposed sensitive and tolerant RAW264.7 cell macrophages (RAW) were stimulated with or without LPS for 4 h or 24 h. Loading control: GAPDH. (**A**) LC3b protein expression was detected in the cytosolic extract by Western blot. (**B**) Western blot image quantification of LC3b-II using image-J software (n = 5 blots; * *p* < 0.05). (**C**) FFA-exposed tolerant RAW cells were treated with SIRT2 inhibitor AK-7 (25 µM) or vehicle (DMSO; equal volume) and incubated further for 20 h and stimulated with or without LPS as indicated. LC3b protein expression was detected in the cytosolic extract by Western blot. (**D**) Western blot image quantification of LC3b-II using image-J software (n = 5 blots; * *p* < 0.05).

**Figure 4 cells-10-00731-f004:**
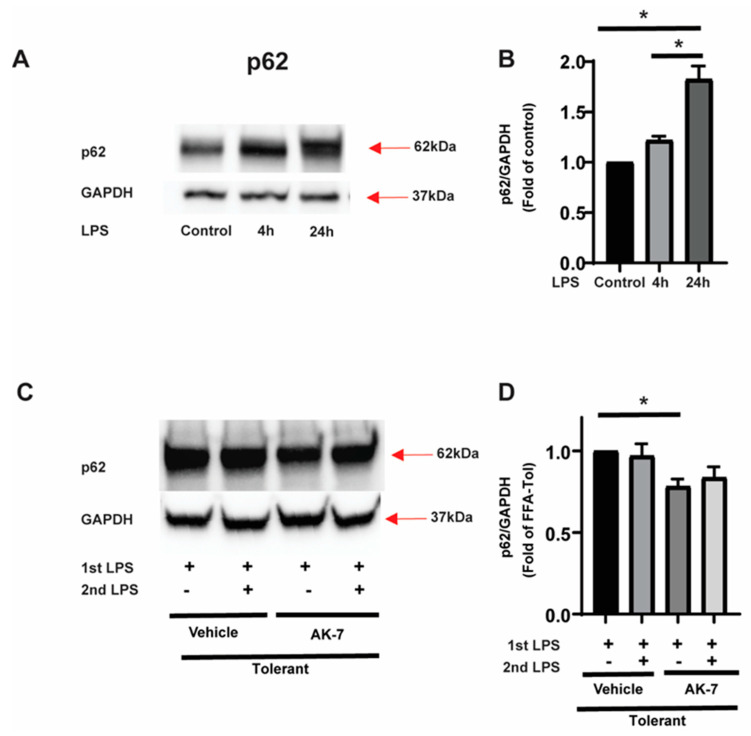
p62 expression in sensitive and tolerant macrophages. Stearic acid (free fatty acid: FFA)-exposed sensitive and tolerant RAW264.7 cell macrophages (RAW) were stimulated with or without LPS for 4 h or 24 h. Loading control: GAPDH. (**A**) p62 protein expression was detected in the cytosolic extract by Western blot. (**B**) Western blot image quantification using image-J software (n = 5 blots; * *p* < 0.05). (**C**) FFA-exposed tolerant RAW cells were treated with SIRT2 inhibitor AK-7 (25 µM) or vehicle (DMSO; equal volume) and incubated further for 20 h and stimulated with or without LPS as indicated. p62 protein expression was detected in the cytosolic extract by Western blot. (**D**) Western blot image quantification of p62 using image-J software (n = 5 blots; * *p* < 0.05).

**Figure 5 cells-10-00731-f005:**
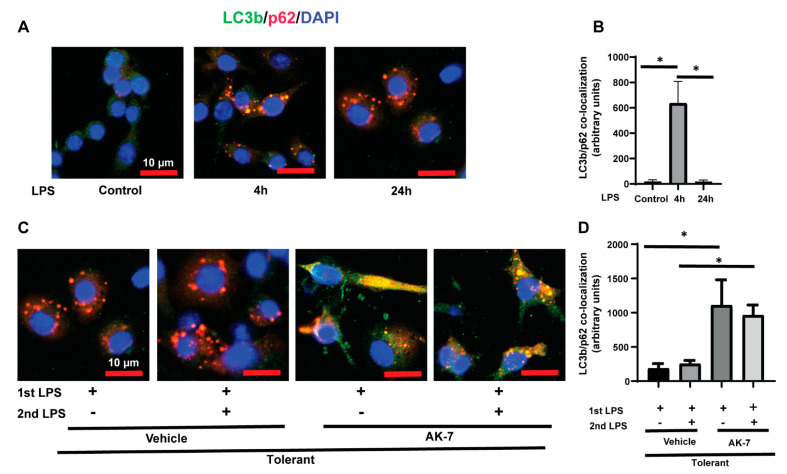
Elevated interaction of LC3b and p62 in sensitive but not tolerant macrophages. Stearic acid (free fatty acid: FFA)-exposed sensitive and tolerant RAW264.7 cell macrophages (RAW) were stimulated with or without LPS as indicated. (**A**) Representative Images of LC3b (green) and p62 (red) double immunostaining (yellow) in FFA-exposed sensitive and tolerant RAW cells following LPS stimulation as indicated. The yellow color indicates colocalization of LC3b and p62. (**B**) Fluorescence quantification of LC3b and p62 double immunostaining (yellow) in FFA-exposed sensitive and tolerant RAW cells. (**C**) FFA-exposed tolerant RAW cells were treated with SIRT2 inhibitor AK-7 (25 µM) or vehicle (DMSO; equal volume) and incubated further for 20 h and stimulated with or without LPS as indicated. Representative Images of LC3b (green) and p62 (red) double immunostaining (yellow) in FFA-exposed sensitive and tolerant RAW cells following LPS stimulation as indicated. (**D**) Fluorescence quantification of LC3b and p62 double immunostaining in FFA-exposed tolerant RAW cells in the presence or absence of AK-7. * *p* < 0.05.

**Figure 6 cells-10-00731-f006:**
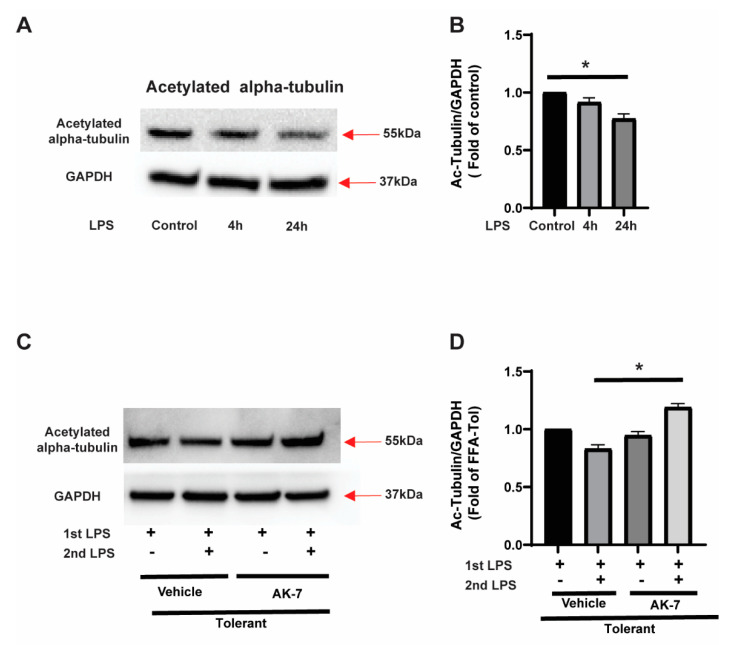
Acetylated alpha-tubulin expression in sensitive and tolerant macrophages. Stearic acid (free fatty acid: FFA)-exposed sensitive and tolerant RAW264.7 cell macrophages (RAW) were stimulated with or without LPS for 4 h or 24 h. Loading control: GAPDH. (**A**) Acetylated alpha-tubulin protein expression was detected in the cytosolic extract by Western blot. (**B**) Western blot image quantification using image-J software (n = 5 blots; * *p* < 0.05). (**C**) FFA-exposed tolerant RAW cells were treated with SIRT2 inhibitor AK-7 (25 µM) or vehicle (DMSO; equal volume) and incubated further for 20 h and stimulated with or without LPS as indicated. Acetylated alpha-tubulin protein expression was detected in the cytosolic extract by Western blot. (**D**) Western blot image quantification of acetylated alpha-tubulin using image-J software (n = 5 blots; * *p* < 0.05).

**Figure 7 cells-10-00731-f007:**
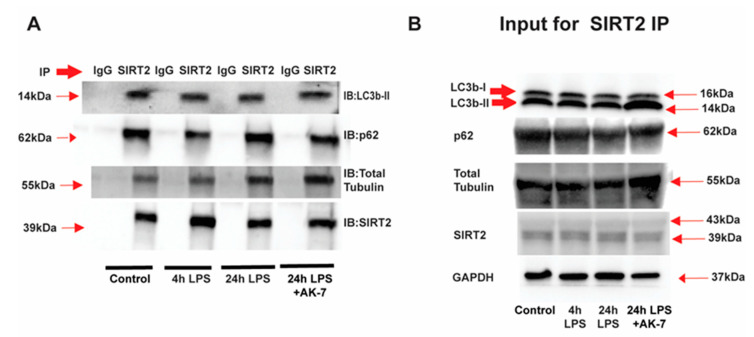
SIRT2 interacts with p62, LC3b and α-tubulin in sensitive and tolerant macrophages. Stearic acid (free fatty acid: FFA)-exposed sensitive and tolerant RAW264.7 cell macrophages (RAW) were stimulated with or without LPS as indicated. FFA-exposed tolerant RAW cells were treated with AK-7 or vehicle (DMSO) and stimulated with or without LPS as indicated. (**A**) SIRT2 was immunoprecipitated (IP) from whole-cell lysates of FFA-exposed sensitive and tolerant RAW cells using an anti-SIRT2 antibody followed by immunoblot (IB) analysis of the indicated proteins including LC3b-II, p62, total alpha-tubulin and SIRT2. Pull down with IgG control antibody was used as a negative control. (**B**) Western blot analysis of LC3b, p62, total alpha-tubulin and SIRT2 in the whole cell lysate used as input for the SIRT2 IP. Loading control: GAPDH.

**Figure 8 cells-10-00731-f008:**
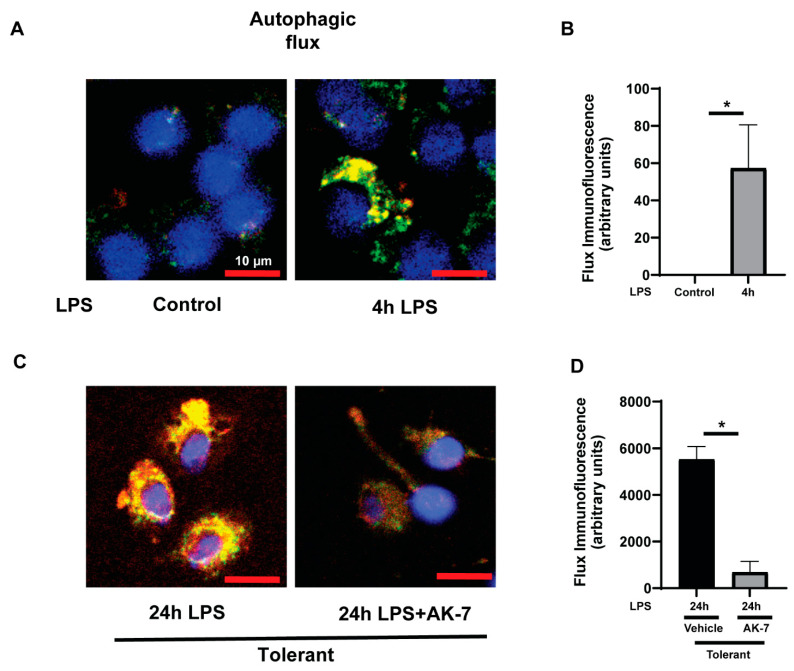
SIRT2 inhibitor AK7 increases autophagic clearance in the tolerant macrophages. Stearic acid (free fatty acid: FFA)-exposed sensitive and tolerant RAW264.7 cell macrophages (RAW) were stimulated with or without LPS as indicated. (**A**) Representative images of autophagosome formation (yellow puncta) with mRFP-GFP-LC3 transduction in FFA-exposed sensitive RAW cells. (**B**) Fluorescence quantification of mRFP/GFP fluorescence in FFA-exposed RAW cells in control (without LPS) and LPS. (**C**) Representative images of autophagosome formation (yellow puncta) with mRFP-GFP-LC3 transduction in FFA-exposed tolerant RAW cells in the presence or absence of AK-7. (**D**) Fluorescence quantification of mRFP/GFP fluorescence in tolerant RAW cells in the presence or absence of AK-7. Autophagosome formation indicated by merged GFP and RFP fluorescence, indicated by yellow-puncta. * *p* < 0.05.

## Data Availability

The data presented in this study are available in [insert Appendix A].

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
