# Peer review of "Sirtuin 2 Dysregulates Autophagy in High-Fat-Exposed Immune-Tolerant Macrophages"

_cells, 2021, doi:10.3390/cells10040731_

Round 1

Reviewer 1 Report

This is an interesting work which is within the scope of the journal. Experiments were logical and the results are presented in a clear manner. I would suggest the authors to double check their Westerns with original data to avoid issues.

Author Response

Thank you for the comment. We have now confirmed the western blots.

Reviewer 2 Report

The results presented by Dr. Vachharajani and colleagues build upon their previous studies showing a role for SIRT enzymes in regulating inflammation. The manuscript is well written with logical experimental flow and the topic of investigation is novel. It is commendable that authors conducted a detailed investigation of the major autophagy related proteins in this preliminary characterization study and also performed immunoprecipitation study to correlate the role of SIRT2 with the major autophagy proteins.

Comments –

  1. Please label the molecular weight markers on all western blot images.
  2. Please provide a rational for the dose and acute time point selected for stearic acid exposure. Were any preliminary dose response or time course studies for FFA exposure conducted?
  3. Obesity is a long term chronic disorder. Please add to the discussion section the relevance of the findings presented with acute exposure of FFA.
  4. Could the authors confirm the effect of FFA exposure on other classical pro-inflammatory cytokine, namely IL-6, at least for the findings in Figure 1A and 1F. This will help strengthen the overall concept.
  5. Please comment on the double bands observed on western blot for SIRT2.
  6. With respect to findings of Figure 1D, did authors also conduct western blot analysis to check if SIRT2 expression is increased in tolerant cells?
  7. Was there any effect of FFA exposure in sensitive and LPS tolerant macrophages on cell survival? Or alternatively does the FFA induced increased inflammatory activation upon LPS stimulation cause any toxicity and does AK-7 protects against that?

Author Response

We thank the reviewer for insightful and thorough review. Please find the response below:

Comment 1: Please label the molecular weight markers on all western blot images.

Response: Done in all figures.

Comment 2: Please provide a rational for the dose and acute time point selected for stearic acid exposure. Were any preliminary dose response or time course studies for FFA exposure conducted?

Response: This is based on our previous work and dose response prior to that: explanation added Section 2.1, line 2. Ref added (ref no. 12).

Comment 3: Obesity is a long term chronic disorder. Please add to the discussion section the relevance of the findings presented with acute exposure of FFA.

Response: Done (page 14, paragraph 1, line1-5)

Comment 4:  Could the authors confirm the effect of FFA exposure on other classical pro-inflammatory cytokine, namely IL-6, at least for the findings in Figure 1A and 1F. This will help strengthen the overall concept.

Response: Done: Figure added (Figure 1E), text added (Page 5, paragraph 2).

Comment 5: Please comment on the double bands observed on western blot for SIRT2.

Response: SIRT2 has two isoforms, as noted in literature. Added to the results, reference added too (page 4, section 3.1, paragraph 2, line:3).

Comment 6: With respect to findings of Figure 1D, did authors also conduct western blot analysis to check if SIRT2 expression is increased in tolerant cells?

Response: We have now added the Western blot to the immunocytochemistry, and both are now moved to supplementary figures 1C and D.

Comment 7: Was there any effect of FFA exposure in sensitive and LPS tolerant macrophages on cell survival? Or alternatively does the FFA induced increased inflammatory activation upon LPS stimulation cause any toxicity and does AK-7 protects against that?

Response: We did not detect FFA-induced toxicity in RAW cells. We clarified this point in the manuscript (page 2, section 2.1, paragraph 1, line 2-5).

Reviewer 3 Report

In this manuscript entitled ‘Sirtuin 2 Dysregulates Autophagy in High-Fat Exposed Immune Tolerant Macrophages’ authors demonstrate the role of sirtuin 2 (SIRT2) in the dysregulation of the autophagic flux in the endotoxin tolerant hypo-inflammatory phases of macrophages.

The results can be better explained and the figure legends can be more explicative.

Please find below a list of comments and concerns that I hope the authors will find helpful in improving this manuscript.

Major points:

  • The authors, to evaluate the role of SIRT2 in the impairment of the autophagic flux, use AK-7 inhibitor. To obtain a stronger result a knock-down experiment of SIRT2 can be done.
  • Figure 3: Following Klionsky guidelines (doi: 1080/15548627.2015.1100356), to evaluate the impairment or the recovery of autophagic flux studying the LC-3 marker, is important to block the last steps of autophagy with chloroquine, bafilomycin or other compounds.
  • Figure 1C: immunostaining of SIRT2 is not really convincible. Try to evaluated SIRT2 expression with DAPI or with a different lens and magnification.

Minor points:

  • The section Materials and methods can be improved and organized well.

Line 86 and 139 change uL whit µL

Line 152 change RAW264.7F with the correct name.

Line 98 and 129 change ‘raw cells’ with the full name of the cell line.

Western blot analysis section: How many micrograms of protein were loaded on the gel?

Doing antibodies section.

  • Is important to insert bars on fluorescence images.

  • Figure 1 B: the better evaluate the difference is necessary to enlarge the western blot figure and explain better the loading of the gel. What are the sensitive one and the tolerant one?

  • Figure 4: p62 expression are overexposed. With a lower exposition the difference can be more appreciable.

  • Figure 5 A (24 hours): is not clear and seems to be obtained with a different magnification. Another images can be more explicative.

  • SQSMT1 in the abbreviation is not correct because is not present in the text.

Author Response

We thank the reviewer for a thorough review and insightful comments. We have answered all of them as outlined below. As a result, we feel that the manuscript quality has now improved tremendously. 

Comment 1: The authors, to evaluate the role of SIRT2 in the impairment of the autophagic flux, use AK-7 inhibitor. To obtain a stronger result a knock-down experiment of SIRT2 can be done.

Response: Thank you for this comment. The goal was to study the effect of SIRT2 induction with the endotoxin tolerance during late/hypo-inflammatory phase. Using SIRT2 KO macrophages, with SIRT2 deficiency at the outset (even during the hyper-inflammation), would not be able to recapitulate the effect of SIRT2 in dysregulating autophagy during late/hypo-inflammation. However, this line of investigation is being pursued in a separate project.

Comment 2: Figure 3: Following Klionsky guidelines (doi: 1080/15548627.2015.1100356), to evaluate the impairment or the recovery of autophagic flux studying the LC-3 marker, is important to block the last steps of autophagy with chloroquine, bafilomycin or other compounds.

Response: This is an important point indeed. We report dysregulation of autophagy during hypo-inflammatory/endotoxin tolerant phase. So, instead, we performed the opposite (to the suggestion) experiment, we induced autophagy using rapamycin. We observed that rapamycin induced autophagy (as expected) in sensitive cells with autophagy clearance (Supplementary figure 4, text: page: 9, paragraph: 2). However, during the endotoxin tolerant phase, there was continued accumulation of LC3 in rapamycin treated cells, indicating failure of clearance of autophagy. This data, when read in conjunction with improved autophagy clearance in AK-7 treated cells, suggests a significant role for SIRT2 in autophagy dysregulation.

Comment 3: Figure 1C: immunostaining of SIRT2 is not really convincible. Try to evaluated SIRT2 expression with DAPI or with a different lens and magnification.

Response: Thank you. Yes, agreed. We now show this better as included in supplementary figure 1A (according to another reviewer suggestion).We also included Western blot images for these groups to corroborate our immunofluorescence data (Supplementary figure 1C).

Minor points:

The section Materials and methods can be improved and organized well.

Comment 4: Line 86 and 139 change uL whit µL

Response: Done (page 3, line 143)

Comment 5: Line 152 change RAW264.7F with the correct name.

Response: Done (Page 3, line 133).

Comment 6: Line 98 and 129 change ‘raw cells’ with the full name of the cell line.

Response: Done (Page 4, section: 3.1, paragraph:2, line: 4).

Comment 7: Western blot analysis section: How many micrograms of protein were loaded on the gel?

Response: added to the methods

Comment 8: Is important to insert bars on fluorescence images.

Response: Done.  Scale bars are included in Figure 5, 8, S1,S2 and S4.

Comment 9: Figure 1 B: the better evaluate the difference is necessary to enlarge the western blot figure and explain better the loading of the gel. What are the sensitive one and the tolerant one?

Response: Agree. We have now moved the immunocytochemistry figure to supplement to allow more space for this figure and have enlarged the western blots, and amount of protein used for loading gels is also specified.

Comment 10: Figure 4: p62 expression are overexposed. With a lower exposition the difference can be more appreciable.

Response: Thank you. Yes, we tried. However, this protein seems to be really abundantly present…

Comment 11: Figure 5 A (24 hours): is not clear and seems to be obtained with a different magnification. Another images can be more explicative.

Response: Thank you. Yes, we redid imaging and added scale bars to bring clarity.

Comment 12: SQSMT1 in the abbreviation is not correct because is not present in the text.

Response: Agreed. Removed.

Reviewer 4 Report

In this manuscript the authors showed impaired autophagosome formation and autophagy clearance via increased SIRT2 expression in FFA‐exposed macrophages. This is a very thorough study, with a lot of data and experiments.

Missing the phrase "in vitro".

However, the authors state that autophagy is induced during the endotoxin sensitive hyper‐inflammatory phase and dysregulated during the endotoxin‐tolerant hypo‐inflammatory phase. In vitro they must add. They should not forget that they are working with an LPS-induced cell model, and not with septic human samples.

Discussion: Moreover, we showed that SIRT2 inhibition using AK‐7 reversed in vivo endotoxin tolerance and improved survival [11]. The authors should make clear that they showed this in a previous syudy, and not the present one. Eg use "we have previously shown".

So my major comment is why didn't the authors choose to isolate macrophages from septic patients?

I understand it is much easier working with cultured cells, but macrophages from patients would provide "real-life" data. Macrophage isolation is quite straight-forward and since you actually mention sepsis in your text, a limitations paragraph is needed. also be more careful when applying your results from your in vitro model in sepsis. Generalizations should be avoided.

Author Response

We thank the reviewer for very insightful comments. All the concerns are now addressed thoroughly. We feel, the manuscript has improved tremendously as a result, so thank you again.

Comment1 : Missing the phrase "in vitro".

However, the authors state that autophagy is induced during the endotoxin sensitive hyper‐inflammatory phase and dysregulated during the endotoxin‐tolerant hypo‐inflammatory phase. In vitro they must add. They should not forget that they are working with an LPS-induced cell model, and not with septic human samples.

Response: Yes, agreed. This point is made clear in several places (Page2, paragraph: 3, last three lines; Discussion: page 14, paragraph 3).

Comment 2: Discussion: Moreover, we showed that SIRT2 inhibition using AK‐7 reversed in vivo endotoxin tolerance and improved survival [11]. The authors should make clear that they showed this in a previous study, and not the present one. Eg use "we have previously shown".

Response: Agreed and clarified (page 14, paragraph: 4, line 7).

Comment 3: So my major comment is why didn't the authors choose to isolate macrophages from septic patients? I understand it is much easier working with cultured cells, but macrophages from patients would provide "real-life" data. Macrophage isolation is quite straight-forward and since you actually mention sepsis in your text, a limitations paragraph is needed. also be more careful when applying your results from your in vitro model in sepsis. Generalizations should be avoided.

Response: Thank you for this comment. The goal of this study was to perform mechanistic studies using in vitro system. Sepsis patients present at different time points during the course of the disease and there are no clear biomarkers to quickly identify the phase of sepsis, making it difficult to study hyper- versus hypo-inflammatory phase in vivo in patients. Thus, the in vitro system for proof-of-concept studies. However, we agree that this is a definite limitation and should be pointed out as such. We have now dedicated a paragraph to point out the limitations and this point is included. In addition, we have added “in vitro” to make it clear that this is an in vitro system, in various places throughout the manuscript as delineated above.

Round 2

Reviewer 2 Report

Authors have addressed all review concerns and manuscript is highly recommended for publication.

Author Response

Thank you for the remarks.

Reviewer 3 Report

The data add in supplementary figure 2 in the treatment with the vehicle support the data but the treatment with rapamycin is not enough. To evaluate the autophagy impairment with LC-3 (the most important marker of autophagic flux) is necessary the use of an inhibitor that blocks the last steps of autophagy to evaluate LC-3II accumulation. 

In supplementary figure 1 the magnification is different between sensitive cells and tolerant cells. 

The protein's micrograms were added in the results. Can be better insert it in materials and methods in western blot section if the protein micrograms in the gel is every time the same. 

The Antibody section in Materials and Methods is missing (line 75-85 is Antibody section).  

Author Response

Comment 1: evaluate the autophagy impairment with LC-3 (the most important marker of autophagic flux) is necessary the use of an inhibitor that blocks the last steps of autophagy to evaluate LC-3II accumulation. 

Response: Thank you for the comment. We have now performed the experiment with chloroquine in FFA-sensitive and tolerant groups and the data appears in Supplementary figure 2. While chloroquine treatment showed increased LC3b-II expression in control and sensitive cells, we did not observe significant change in FFA-tolerant cells with chloroquine vs. without. We can explain this finding based on the fact that there are significant dysregulations in autophagy including Beclin-1. 

Comment 2: In supplementary figure 1 the magnification is different between sensitive cells and tolerant cells.

Response: The scale bars are included. We agree that the tolerant cells appear bigger in size, but it is our observation that the tolerant cells appear bigger in size. While out of scope for this manuscript, this phenomenon needs further evaluation.

Comment 3: The protein's micrograms were added in the results. Can be better insert it in materials and methods in western blot section if the protein micrograms in the gel is every time the same. 

Response: Thank you. This is now inserted in the methods (section 2.6; line 3).

Comment 4: The Antibody section in Materials and Methods is missing (line 75-85 is Antibody section).

Response: Thank you. Yes, now labeled as such. 

Reviewer 4 Report

All comments have been addressed.

Author Response

Thank you

Round 3

Reviewer 3 Report

The changes are sufficient to be accepted.